# ViTacFormer: Learning Cross-Modal Representation for Visuo-Tactile Dexterous Manipulation

**Abstract:** Dexterous manipulation is crucial for robots to interact with the physical world. While vision-based methods have advanced rapidly, tactile sensing remains essential for fine-grained control, especially under occlusion. We present ViTacFormer, a cross-modal framework that fuses vision and touch via cross-attention and predicts future tactile states with an autoregressive head. A curriculum gradually shifts from ground-truth to predicted tactile inputs, stabilizing representation learning. On real-world benchmarks covering both short- and long-horizon tasks, ViTacFormer improves success rates by about 50% over strong baselines, and is the first to complete 11-stage dexterous manipulation with 2.5 minutes of continuous operation.

**Keywords:** Dexterous Manipulation, Visuo-Tactile Fusion, Cross-Attention, Autoregressive Tactile Forecasting, Imitation Learning

## 1 Introduction

Recent years have witnessed rapid advances in robotic manipulation [1, 2, 3, 4, 5, 6, 7, 8, 9, 10], with behavior cloning (BC) [11, 12, 13] emerging as a promising paradigm for high-precision real-world tasks. However, most existing approaches remain restricted to simple hand configurations [14] and generalize poorly—largely due to the underutilization of tactile sensing [15, 16, 17, 18], which is indispensable for fine-grained control.

While some studies have attempted to integrate tactile feedback into dexterous manipulation [19, 20], the learned tactile features are often shallow and underexplored. Self-supervised methods have also been applied to tactile data [21, 22, 23, 24, 25], but a unified cross-modal representation for visuo-tactile dexterous manipulation remains missing [26, 27].

We introduce **ViTacFormer**, a visuo-tactile framework that addresses this gap. Our key idea is to fuse high-resolution vision and tactile cues with cross-attention layers at every stage of the policy, and to enforce predictive modeling of future tactile states. This tactile-prediction head drives the latent space to encode actionable touch dynamics, providing richer cues than perceiving only current signals.

Since autoregressive tactile forecasting is inherently challenging, we design a **two-phase curriculum**: during the first 75% of training, ground-truth tactile signals stabilize learning; during the final 25%, predicted signals are used to promote robust cross-modal reasoning.

To evaluate ViTacFormer, we construct the first comprehensive real-world benchmark for visuo-tactile dexterous manipulation, covering four short-horizon tasks and a very long-horizon, 11-stage task. Across all benchmarks, ViTacFormer achieves about **50%** higher success rates than strong baselines, and is the first system to complete continuous dexterous operation for **2.5 minutes** over **11 sequential stages**.

In summary, our contributions include:

- A real-world experimental setup with bi-manual anthropomorphic hands, teleoperation, and a benchmark for visuo-tactile dexterous manipulation.
- A multimodal learning framework that couples cross-attention fusion with autoregressive tactile prediction and curriculum training.
- Strong empirical results, including significant gains over baselines and the first successful demonstration of long-horizon dexterous manipulation on a real robot.

## 2 Problem Formulation and Hardware Setup

### 2.1 Problem Formulation

We study imitation learning for dexterous bi-manual manipulation. Given a set of expert trajectories $\mathcal{D} = \{\tau_i\}_{i=1}^N$, where each $\tau_i = \{(o_t^i, a_t^i)\}_{t=1}^{T_i}$ consists of multimodal observations $o_t^i$ and corresponding actions $a_t^i$, the goal is to learn a policy $\pi_\theta$ that maps observations to actions, $a_t = \pi_\theta(o_t)$. The policy is trained to imitate expert behavior and evaluated on both short- and long-horizon manipulation tasks.

### 2.2 Hardware Setup

Our platform (Fig. 1) consists of two robot arms with anthropomorphic dexterous hands, equipped with wrist-mounted and stereo cameras for vision and fingertip sensors for touch. We collect demonstrations via a custom teleoperation system where operators use exoskeleton gloves and a VR interface with real-time visual–tactile feedback. This setup enables intuitive control and provides high-quality multimodal trajectories.

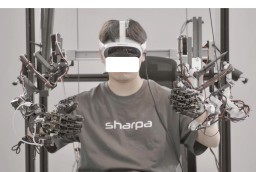

Figure 1: System overview. (a) Dual-arm platform with dexterous hands, wrist cameras, and stereo camera. (b) Teleoperation with exoskeleton gloves and VR headset. (c) VR interface with visual and tactile feedback.

## 3 Method

In section 3.1, we introduce a cross-attention-based multimodal integration framework that fuses the visual and tactile inputs. In section 3.2, we present autoregressive modeling with tactile forecasting, which generates actions conditioned on predicted tactile signals. In section 3.3, we summarize the overall learning procedure for ViTacFormer.

### 3.1 Cross-Attention-Based Multimodal Integration

Visual observations and tactile signals provide complementary cues but naive fusion fails to capture their correlations. Cross-attention enables queries from one modality to attend to keys and values from the other, allowing ViTacFormer to extract semantically relevant dependencies between vision and touch. The resulting features are concatenated into hidden states for downstream action generation.

### 3.2 Auto-Regressive Modeling with Tactile Signal Forecasting

Beyond perceiving current signals, anticipating future tactile feedback improves robustness. ViTac-Former predicts future tactile tokens using $z$, proprioception, and visuo-tactile observations, and concatenates them with current inputs for action generation.

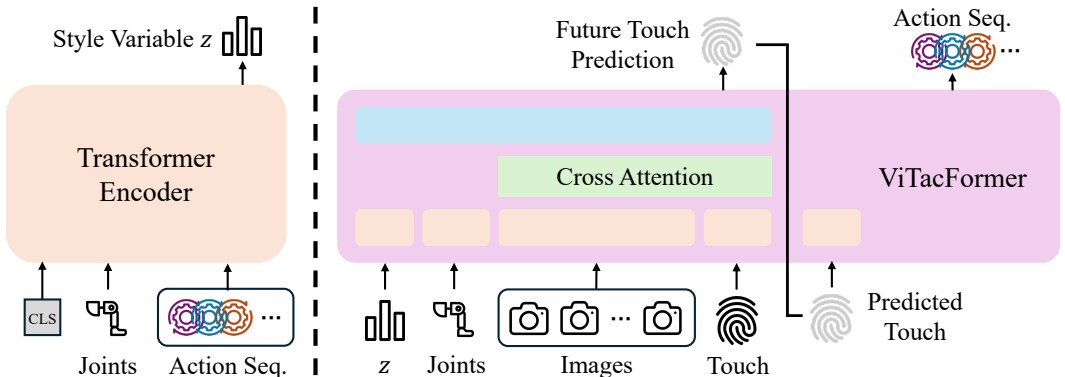

Figure 2: ViTacFormer architecture. A transformer-based encoder maps action sequence and robot proprioception to a style variable $z$. A transformer-based encoder-decoder fuses $z$ with visuo-tactile observations to autoregressively predict future tactile signals and generate actions.

## 3.3 Neural Network Architecture and Learning Procedure

Fig. 2 illustrates the architecture of ViTacFormer, formulated as a conditional variational auto-encoder. On the left, a transformer-based encoder maps the robot's proprioception and expert action sequence into a latent style variable $z$. On the right, a transformer-based encoder-decoder integrates $z$ with visuo-tactile observations via cross-attention, predicts future tactile signals autoregressively, and generates actions accordingly. During training, $z$ is sampled from expert demonstrations, while during inference it is fixed to zero, ensuring consistent action generation without stochastic variation.

Since autoregressive tactile forecasting is challenging at the start of training, we adopt a two-phase curriculum: the first 75% of epochs use ground-truth tactile signals to stabilize learning, and the final 25% gradually switch to predicted signals, encouraging robust cross-modal reasoning under realistic conditions [28].

## 4 Experiment

We evaluate ViTacFormer on four short-horizon dexterous tasks (Fig. 3) and one long-horizon task (making hamburgers). Each task uses only 50 demonstrations, making the setting challenging. We compare against Diffusion Policy (DP) [29], HATO [20], ACT [30], and ACTw/T [30]. DP and ACT do not use tactile inputs, while HATO and ACTw/T fuse tactile signals naively.

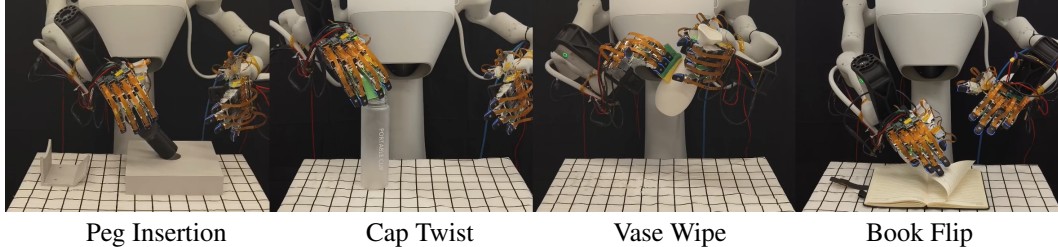

| Peg Insertion | Cap Twist | Vase Wipe | Book Flip |

Figure 3: Four short-horizon visuo-tactile tasks, from left to right, i.e., peg insertion, cap twist, vase wipe, and book flip.

## 4.1 Algorithm Comparison

To evaluate the effectiveness of our approach, we conduct experiments on four short-horizon dexterous manipulation tasks, with each algorithm tested for 10 rollouts per task. As shown in Tab. 1,

Table 1: Success rate comparison on four short-horizon dexterous manipulation tasks. Our ViTac-Former achieves over $50\%$ success rates compared to the baselines.

| Task | Peg Insertion | Cap Twist | Vase Wipe | Book Flip |
|---|---|---|---|---|
| DP [29] | 2/10 | 0/10 | 3/10 | 1/10 |
| ACT [30] | 4/10 | 4/10 | 3/10 | 2/10 |
| HATO [20] | 4/10 | 1/10 | 4/10 | 3/10 |
| ACTw/T [30] | 6/10 | 6/10 | 4/10 | 4/10 |
| **Ours** | **10**/10 | **10**/10 | **9**/10 | **9**/10 |

ViTacFormer consistently outperforms all baselines, achieving markedly higher success rates and nearly solving these challenging benchmarks.

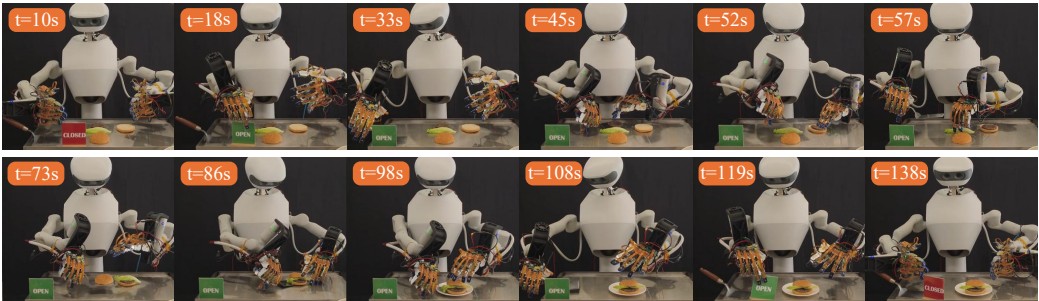

Figure 4: Successful rollout of ViTacFormer on the long-horizon hamburger task.

We further evaluate ViTacFormer on a challenging 11-stage long-horizon task of making hamburgers. As shown in Fig. 4, ViTacFormer is the first to complete the full sequence on a real robot, demonstrating stable performance across all stages and achieving over 80% overall success rate.

## 4.2 Ablation Study

Table 2: Ablation study on four short-horizon dexterous tasks. Each proposed component improves performance over the baseline.

| Task | Peg Insertion | Cap Twist | Vase Wipe | Book Flip |
|---|---|---|---|---|
| w/o Tactile | 4/10 | 4/10 | 3/10 | 2/10 |
| w/o CrossAttention | 9/10 | 7/10 | 7/10 | 7/10 |
| w/o AutoRegressive | 7/10 | 7/10 | 6/10 | 7/10 |
| w/o Two-Stage | 7/10 | 6/10 | 4/10 | 6/10 |
| **Ours** | **10/10** | **10/10** | **9/10** | **9/10** |

Results in Tab. 2 show that removing any component leads to noticeable performance drops. Cross-attention enhances vision-touch fusion, autoregressive tactile forecasting improves stability, and the two-stage curriculum stabilizes training. Together, these components enable ViTacFormer to achieve the best performance.

## 5 Conclusion

We present ViTacFormer, a visuo-tactile framework for dexterous manipulation that integrates cross-modal fusion and predictive tactile modeling. With a curriculum strategy to stabilize training, ViTacFormer achieves robust control and outperforms strong baselines by large margins. Notably, it is the first system to complete long-horizon dexterous tasks on a real robot, highlighting the potential of vision-touch integration for generalizable manipulation.

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

# Appendix

## A    Additional Method Details

### A.1    Input Modalities

Our model takes multimodal inputs from the robot system, including visual observations, robot proprioception, and tactile signals.

**Visual Input**

We use four synchronized camera views as visual input: a stereo pair (180×320) from top-mounted ZED Mini cameras (Fig. 5(**a**), (**c**)), and two fisheye wrist-mounted views (256×280) for left and right hands (Fig. 5(**b**), (**d**)). All frames are encoded into image tokens via a vision backbone before cross-modal integration.

**Proprioception Input**

The robot's internal state at each timestep is represented by a 58-dimensional vector, consisting of: 7-DoF left arm state, 17-DoF left hand state, 7-DoF right arm state, 17-DoF right hand state, and 2-DoF neck state—structured as $[7, 17, 7, 17, 2]$. A temporal horizon of 6 frames is used, resulting in a proprioceptive input of shape $(6, 50)$.

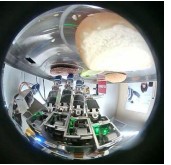

(a) Left eye          (b) Left wrist

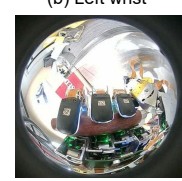

(c) Right eye          (d) Right wrist

Figure 5: **Four types of camera views**

**Tactile Input**

Each of the 10 fingertips is equipped with force and torque sensors along 3 axes, resulting in 20 tactile channels. For each channel, we collect 18 frames of data ($[18, 3]$), which are concatenated into a raw tactile tensor of shape $[18, 60]$. We additionally compute frame-wise deltas to obtain relative changes ($[18, 60]$), and concatenate them with the raw signal to produce the final tactile input of shape $[18, 120]$.

### A.2    Action Output

The policy generates high-frequency action sequences with shape $(100, 50)$ per rollout, where 50 corresponds to the full control dimension of the robot: 7-DoF left arm, 17-DoF left hand, 7-DoF right arm,17-DoF right hand, and 2-DoF neck—matching the structure of the proprioceptive state. The 100-frame horizon supports fine-grained dexterous motion across extended manipulation stages.

### A.3    Data and training details

We train each task using 50 expert demonstrations and 100 epochs on 2 NVIDIA H20 GPUs. Short-horizon tasks typically converge within half a day, while long-horizon tasks (e.g., Make Hamburger) require up to 2 days. The model is optimized using the Adam optimizer with a learning rate of 1e-4 and a batch size of 128. Training supervision includes KL divergence on latent action style, L1 losses on both predicted actions and tactile signals, and auxiliary supervision on end-effector positions and rotations. All input modalities are temporally aligned and normalized prior to training.

## A.4  Inference Details

During deployment, the policy runs at 10Hz, producing a 100-frame $(100, 50)$ high-frequency action sequence at each decision step. To ensure smooth and physically stable execution, we apply temporal smoothing over the predicted action trajectory before sending commands to the robot. The system is deployed on a real dual-arm platform with synchronized visuo-tactile observation streams and low-latency control.

# B  Additional Experiment Details

## B.1  Evaluation Metrics

In addition to success rates, we also define Human Normalized Score (HNS) to measure dexterous manipulation performance in long-horizon tasks. Success rates alone may not fully capture the quality of execution, especially in contact-rich multi-stage manipulation.

HNS provides a stage-wise evaluation: each task is divided into $N$ stages, and each stage is assigned a raw score $s_i \in \{0, 1, 2, 3\}$ reflecting execution quality. Each stage also has a weight $w_i$ corresponding to its tactile reliance. The overall HNS is then defined as:

$$\text{HNS} = \frac{\sum_{i=1}^{N} w_i \cdot s_i}{3 \cdot \sum_{i=1}^{N} w_i}. \tag{1}$$

This normalized score enables fine-grained comparison across stages while accounting for tactile dependence.

## B.2  Short-horizon tasks

The four short-horizon tasks share a standardized tabletop workspace and a common set of objects, as shown in Fig. 6(a). The workspace is discretized using a printed grid (5cm per square), with the top-left corner defined as the origin $(0, 0)$, as illustrated in Fig. 6(b). During training, each object is placed at a designated grid coordinate. For generalization, we randomly perturb the object's position within a circular region of half-grid radius (i.e., 2.5cm) around its original anchor point.

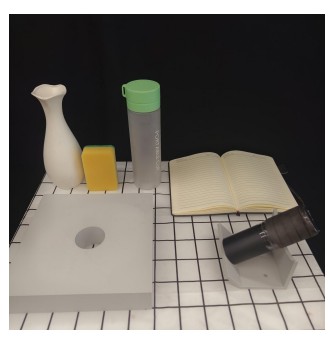
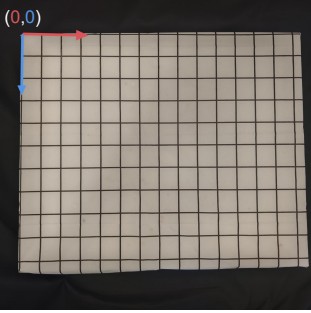

(a) Objects          (b) Workspace

Figure 6: **Short-horizon task setup.** (a) All four short-horizon tasks share a common set of objects. (b) The tabletop workspace is marked with a grid; the top-left corner is defined as the origin $(0, 0)$. Each object is positioned at a predefined grid point during training.

### B.2.1  Peg Insertion

**Task Description**

The robot uses its right hand to grasp a cylindrical peg from the vertical rack, then moves it diagonally along the sloped platform toward the insertion hole. Upon reaching the vicinity of the hole, the robot is expected to insert the peg smoothly and stably into the hole. This task involves visual alignment, precise grasping, and tactile-guided insertion. Representative execution frames are shown in the first row of Fig. 7.

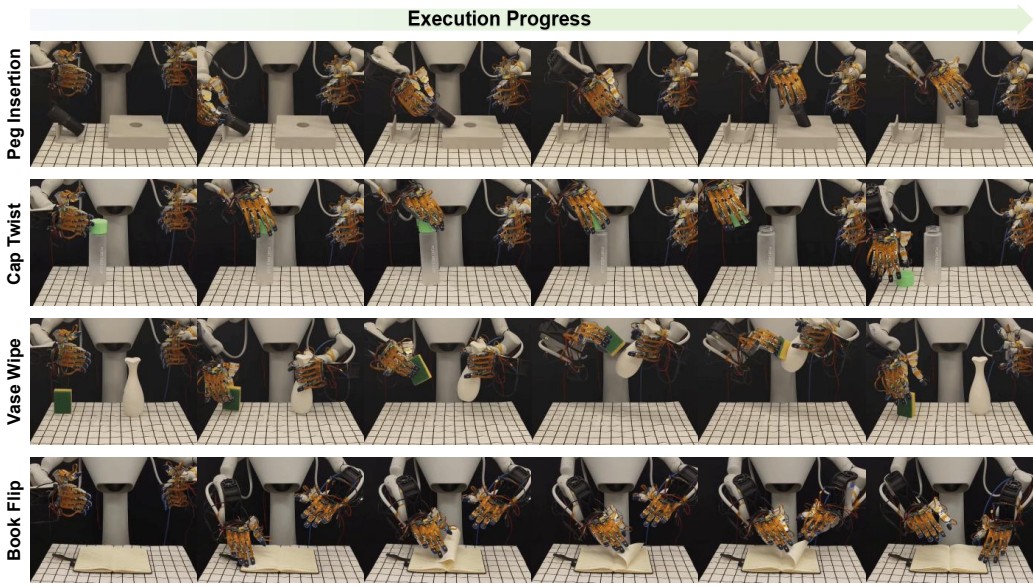

Figure 7: **Execution examples for short-horizon tasks.** Representative keyframes from four tasks: peg insertion, cap twist, vase wipe, and book flip. Each task demonstrates a full execution sequence from perception to manipulation.

**Scoring Scheme**

Table 3: **Scoring criteria for Peg Insertion.**

| Stage 1: Grasp (weight 1) | Description |
| --- | --- |
| 0 | No grasp |
| 1 | Grasped but slipped or dropped |
| 2 | Poor or tilted grasp |
| 3 | Stable grasp |
| **Stage 2: Insertion (weight 2)** | **Description** |
| 0 | No insertion |
| 1 | Misaligned, dropped |
| 2 | Partial insertion |
| 3 | Fully inserted |

The task is divided into two stages: peg grasping (weight 1) and insertion (weight 2). Each stage is scored from 0 to 3 based on qualitative criteria such as grasp stability and insertion completeness. The human normalized score (HNS) is computed as a weighted average. A total score of 3 for stage 1 and $\geq 2$ for stage 2 is considered successful.

**Inference Results**

Table 4 summarizes the quantitative performance on the peg insertion task. We report the average stage-wise scores, human normalized score (HNS), and success rate across baselines and ablations. Our method achieves the highest HNS (0.93) and 100% success rate, demonstrating strong performance across both stages.

Table 4: **Peg Insertion: inference results across models.**

| Model | Stage 1 | Stage 2 | HNS | Success Rate |
|---|---|---|---|---|
| DP | 1.6 | 0.9 | 0.37 | 20% |
| ACT | 2.6 | 1.1 | 0.53 | 40% |
| HATO | 2.4 | 1.1 | 0.51 | 40% |
| ACT w/T | 2.6 | 1.8 | 0.68 | 60% |
| ACT w/CrossAttention | 3.0 | 2.1 | 0.80 | 70% |
| ACT w/NextTouchPred | 2.7 | 2.4 | 0.83 | 80% |
| ACT w/AutoRegressive | 2.9 | 2.2 | 0.81 | 90% |
| Ours | 3.0 | 2.7 | 0.93 | 100% |

**Failure Case Analysis**

Figure 8, first row, shows two representative failure cases in the peg insertion task. In the first case, the robot fails to locate the insertion hole accurately and attempts to insert the peg at an incorrect position, leading to task failure despite a seemingly stable grasp. In the second case, the robot grasps the cylindrical peg with an imprecise hand posture, causing the thumb to slip during the transport phase. As a result, the peg deviates from the planned trajectory and misses the hole entirely.

### B.2.2 Cap Twist

**Task Description**

The robot uses its right hand to rotate a cap off a bottle and place it on the table. The cap is initially tightened at a clockwise offset of about 100 degrees from the open position. Representative execution frames are shown in the second row of Fig. 7.

**Scoring Scheme**

Table 5: **Scoring criteria for Cap Twist.**

| Stage 1: Rotate (weight 2) | Description |
|---|---|
| 0 | No contact with the cap |
| 1 | Rotated 0–50° |
| 2 | Rotated 50–100°, or over-rotated |
| 3 | Fully unscrewed, cap held securely |
| **Stage 2: Place (weight 2)** | **Description** |
| 0 | Dropped immediately or stuck on bottle |
| 1 | Released before full separation |
| 2 | Partially placed or fell off |
| 3 | Stably placed on the table |

The task is divided into two stages: rotation and placement. Each is scored from 0 to 3, and a task is considered successful if the cap is fully unscrewed and placed stably (stage 1 score 3, stage 2 $\geq$2).

**Inference Results**

Table 6 presents the model performance on the cap twist task. Our method achieves the best HNS score (0.98) and 100% success rate, highlighting the advantage of fine-grained tactile reasoning.

**Failure Case Analysis**

In the second row of Fig. 8, two failure cases from the cap twist task are shown. In the first case, the robot fails to detect that the cap has already loosened and continues to apply torque unnecessarily, resulting in over-rotation that destabilizes the object. In the second case, the fingers lose contact

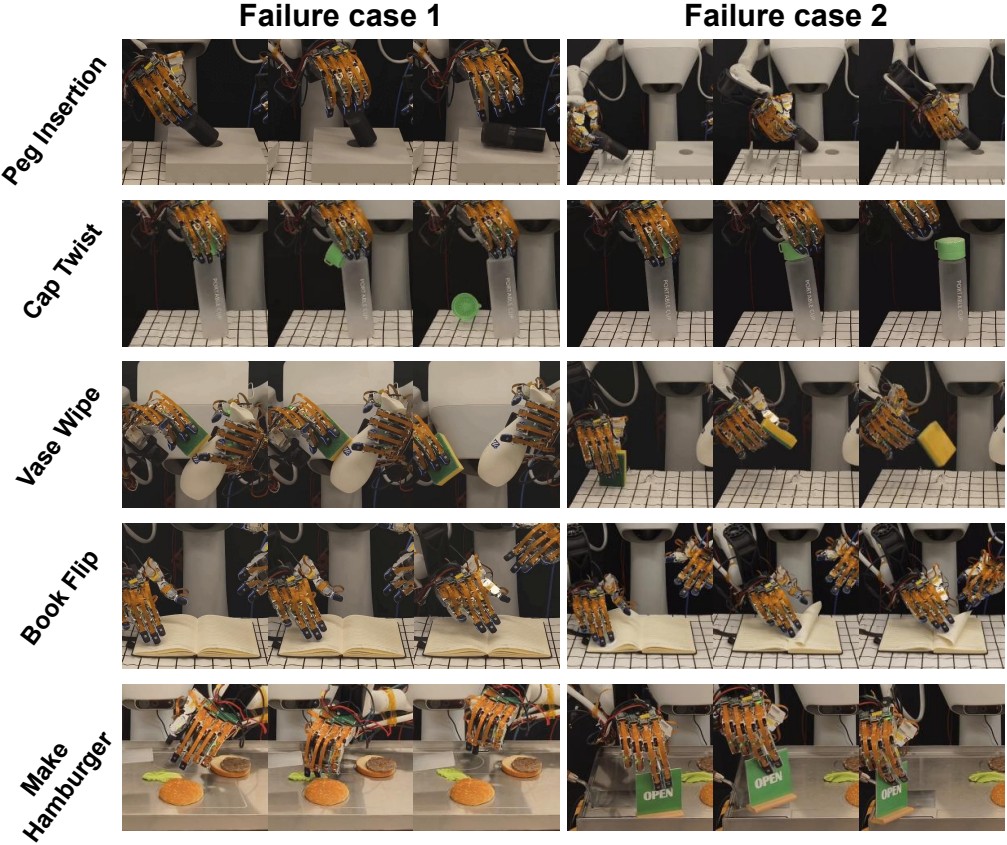

Figure 8: **Representative failure cases across all tasks.** Each row corresponds to one task, with two failure case sequences shown side by side.

Table 6: **Cap Twist: inference results across models.**

| Model | Stage 1 | Stage 2 | HNS | Success Rate |
|---|---|---|---|---|
| DP | 1.1 | 0.3 | 0.23 | 0% |
| ACT | 2.4 | 1.1 | 0.58 | 40% |
| HATO | 1.8 | 0.5 | 0.38 | 10% |
| ACT w/T | 2.6 | 1.8 | 0.73 | 60% |
| ACT w/CrossAttention | 3.0 | 2.3 | 0.88 | 70% |
| ACT w/NextTouchPred | 2.8 | 2.0 | 0.80 | 60% |
| ACT w/AutoRegressive | 2.9 | 2.2 | 0.85 | 70% |
| Ours | 3.0 | 2.9 | 0.98 | 100% |

during the twisting motion, leading to slippage and an insufficient rotation angle, which prevents the cap from being successfully removed.

### B.2.3   Vase Wipe

**Task Description**

The robot uses its left hand to pick up a vase and its right hand to grasp a sponge. It then wipes away the blue ink mark located at the center of the vase. Representative execution frames are shown in the third row of Fig. 7.

**Scoring Scheme**

Table 7: **Scoring criteria for Vase Wipe.**

| Stage 1: Pick (weight 1) | Description |
|---|---|
| 0 | Failed to grasp the sponge |
| 1 | Grasped only a corner of sponge |
| 2 | Unstable grasp with partial control |
| 3 | Firm 3-finger grasp with full control |
| **Stage 2: Wipe (weight 2)** | **Description** |
| 0 | No contact with the ink mark |
| 1 | Wiped less than 50% |
| 2 | Wiped 50–90%, some ink remains |
| 3 | Fully wiped the ink area clean |

The task is divided into two stages: sponge grasping (pick) and vase wiping (wipe), both scored from 0 to 3. If the operator intervenes to re-adjust the vase grasp during stage 1, the score is reduced by 1. The task is considered successful only if both stages score 3.

**Inference Results**

Table 8: **Vase Wipe: inference results across models.**

| Model | Stage 1 | Stage 2 | HNS | Success Rate |
|---|---|---|---|---|
| DP | 1.8 | 1.3 | 0.49 | 30% |
| ACT | 2.0 | 1.5 | 0.56 | 30% |
| HATO | 2.5 | 1.7 | 0.65 | 40% |
| ACT w/T | 3.0 | 1.9 | 0.75 | 40% |
| ACT w/CrossAttention | 3.0 | 2.5 | 0.89 | 60% |
| ACT w/NextTouchPred | 3.0 | 2.2 | 0.82 | 40% |
| ACT w/AutoRegressive | 3.0 | 2.5 | 0.89 | 70% |
| Ours | 3.0 | 2.9 | 0.98 | 90% |

Table 8 shows the quantitative performance on the vase wiping task. Our method again achieves the best HNS (0.98) and 90% success rate, showing reliable grasping and contact-driven wiping.

**Failure Case Analysis**

The third row of Fig. 8 illustrates two typical failure modes in the vase wiping task. In the first case, the robot applies insufficient force during the wiping motion, resulting in incomplete surface contact between the sponge and the vase. Consequently, the ink mark is not fully removed. In the second case, excessive force is applied during the grasping phase, causing the sponge to slip out of the robot's fingers before the wiping action begins.

### B.2.4   Book Flip

**Task Description**

The robot uses its right-hand middle finger to flip up a single page and then presses the page down using its left hand. Representative execution frames are shown in the fourth row of Fig. 7.

**Scoring Scheme**

This task includes two stages: flipping and pressing. Each stage is scored from 0 to 3. The task is considered successful if stage 1 scores 3 and stage 2 scores ≥2.

**Inference Results**

Table 10 shows performance on the book flip task. Our method achieves the highest HNS (0.93) and 90% success rate, outperforming all ablations.

Table 9: **Scoring criteria for Book Flip.**

| Stage 1: Flip (weight 2) | Description |
|---|---|
| 0 | No contact with the page |
| 1 | Touched but failed to lift / flipped multiple pages |
| 2 | Lifted halfway but stopped |
| 3 | Fully flipped one page |

| Stage 2: Press (weight 2) | Description |
|---|---|
| 0 | No contact with the page |
| 1 | Insufficient force, page rebounds |
| 2 | Pressed down, but misaligned |
| 3 | Fully and correctly pressed the page down |

Table 10: **Book Flip: inference results across models.**

| Model | Stage 1 | Stage 2 | HNS | Success Rate |
|---|---|---|---|---|
| DP | 1.5 | 0.5 | 0.35 | 10% |
| ACT | 1.9 | 0.7 | 0.43 | 20% |
| HATO | 2.0 | 0.6 | 0.43 | 30% |
| ACT w/T | 2.3 | 0.9 | 0.53 | 40% |
| ACT w/CrossAttention | 2.7 | 1.9 | 0.77 | 70% |
| ACT w/NextTouchPred | 2.9 | 1.9 | 0.80 | 70% |
| ACT w/AutoRegressive | 2.7 | 2.1 | 0.80 | 70% |
| Ours | 3.0 | 2.6 | 0.93 | 90% |

**Failure Case Analysis**

Figure 8, fourth row, presents two failure modes in the book flip task. In the first case, the robot fails to perceive the presence or precise location of the page edge, resulting in a poking motion that completely misses the page during the flipping attempt. In the second case, the robot applies excessive downward force before initiating the flip, which presses the page flat against the book and prevents it from being lifted.

## B.3 Long-horizon task: Make Hamburger

**Workspace Setup**

The long-horizon task is conducted on a customized metallic tabletop with seven designated ingredient/tool zones, as shown in Fig. 9. Each object is placed within either a circular or rectangular region marked on the tray. These regions serve as initialization zones with controlled spatial variability to support generalization. During both training and evaluation, each item is placed randomly within its assigned zone (up to 3cm positional jitter), ensuring that the policy must perform robust multimodal perception and execution.

**Task Description**

The long-horizon task involves a full hamburger assembly sequence requiring precise tool use and multi-stage coordination. The robot begins by flipping a wooden card from "closed" to

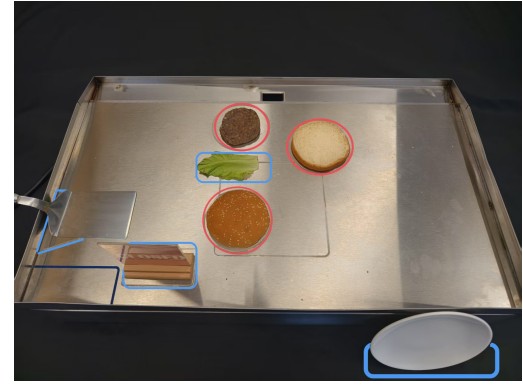

Figure 9: **Long-horizon task setup.** Seven components are placed in predefined zones—circular (ingredients) or rectangular (tools). Objects are randomly initialized within these areas to test spatial generalization.

"open" to indicate the start of service. It then uses its right hand to grasp a spatula and sequentially completes the following steps: (1) lift and place the meat patty onto the bottom bread, (2) place a piece of lettuce, and (3) lift and place the top bread. Once the hamburger is assembled, the robot places it onto a plate handed over by a human. Finally, it returns the spatula to its original position and flips the sign back to "closed" to indicate task completion.

**Scoring Scheme**

The long-horizon hamburger task is decomposed into 11 sequential stages, covering symbolic interaction (sign flipping), tool use (spatula manipulation), ingredient assembly (meat patty, lettuce, bun), and final delivery. Each stage is scored from 0 to 3, where 0 indicates failure or no attempt, 1–2 denote partial or unstable execution, and 3 represents correct and stable completion. To better reflect task complexity and tactile sensitivity, each stage is assigned a specific weight: for example, sign flipping and deformable object handling (lettuce, bun) are given higher weights due to their reliance on fine-grained control and multi-finger dexterity.

The weighted stage scores are used to compute a Human Normalized Score (HNS), which reflects the overall task performance. A stage is considered successful if the score is at least 1. The entire task is marked as successful only when all 11 stages meet this threshold. Table 11 details the scoring criteria and weights for each stage.

Table 11: **Scoring criteria for the long-horizon hamburger task.**

| Stage | Action | Weight | Score Description |
| --- | --- | --- | --- |
| 1 | Flip sign (start) | 2 | 0: miss/fail; 1–2: partial (0–180°); 3: clean flip |
| 2 | Grab spatula | 2 | 0: miss; 1–2: unstable grasp; 3: secure grasp |
| 3 | Lift meat patty | 1 | 0: failed; 1–2: partial lift; 3: stable lift |
| 4 | Place meat patty | 1 | 0: miss; 1–2: partial/inaccurate; 3: centered |
| 5 | Grasp lettuce | 2 | 0: miss; 1–2: loose grasp; 3: stable placement |
| 6 | Lift top bread | 2 | 0: failed; 1–2: unstable or too forceful; 3: correct |
| 7 | Place top bread | 1 | 0: miss; 1–2: inaccurate; 3: clean stack |
| 8 | Lift hamburger | 1 | 0: failed; 1–2: unstable; 3: correct lift |
| 9 | Place on plate | 1 | 0: miss; 1–2: off-center; 3: perfect placement |
| 10 | Return spatula | 1 | 0: drop/fail; 1–2: misaligned; 3: accurate return |
| 11 | Flip sign (end) | 2 | 0: fail; 1–2: partial rotation; 3: clean close flip |

**Failure Case Analysis**

The fifth row of Fig. 8 shows two failure cases from the long-horizon hamburger assembly task. In the first case, the robot fails during stage 5 (grasping the lettuce): the grasp is unstable and incomplete, resulting in the lettuce slipping from the fingers before it can be placed. In the second case, the failure occurs in stage 1 (flipping the sign): although the sign is flipped, an incorrect grasp orientation causes the sign to rotate unintentionally during the movement, leading to a collision with the edge of the stove and blocking task progression.

