# OpenReview forum: "ViTacFormer: Learning Cross-Modal Representation for Visuo-Tactile Dexterous Manipulation"
_robot-learning.org/CoRL/2025/Workshop/Dexterous_Manipulation — CoRL 2025 Workshop Dexterous Manipulation Spotlight_

### Official Review · Reviewer_cwPn · 2025-09-06
**Review of ViTacFormer**

**Rating:** 7
**Confidence:** 4

**Review:**

## Summary
The paper introduces ViTacFormer, a framework for dexterous robotic manipulation that tightly couples vision and tactile sensing. The method employs cross-attention fusion to integrate modalities and an autoregressive tactile forecasting head to anticipate future tactile states. To stabilize the training, the authors propose a two-phase curriculum that transitions from using ground-truth tactile signals to predicted ones. The framework is tested on both short and long tasks, showing a significant improvent.

## Strength
1. Novel Cross-Modal Fusion: The use of cross-attention at all policy stages provides a principled way to align and exploit visual and tactile cues.
2. Predictive Tactile Predicting: Introducing autoregressive tactile forecasting is innovative. It forces the latent space to encode touch dynamics rather than merely current signals.

## Weakness
1. Limited Baseline Diversity: While several methods are compared, most baselines either ignore tactile inputs or fuse them naively. More recent multimodal architectures are not considered.
2. Architectural Complexity: The model combines transformers, cross-attention, a conditional VAE, and autoregressive forecasting. While effective, the computational and deployment overhead may hinder accessibility and real-time applications on resource-constrained systems.

---

### Official Review · Reviewer_BDxJ · 2025-09-09
**review of ViTacFormer**

**Rating:** 8
**Confidence:** 5

**Review:**

This paper proposes a novel tactile-visual representation learning method. Lots of challenging tasks convince me that this technique is powerful.

---

### Decision · Program_Chairs · 2025-09-18

Accept (Spotlight)